I-Cubid: a nonlinear cubic graph-based approach to visualize and in-depth browse Flickr image results

Rashid Umer 1
Saddal Maha 1
Khan Abdur Rehman arkhan@cs.qau.edu.pk 1
Manzoor Sadia 1
Ahmad Naveed 2
1 Department of Computer Sciences, Quaid-i-Azam University , Islamabad , Pakistan
2 College of Computer and Information Sciences, Prince Sultan University , Riyadh , Saudi Arabia
Thompson Steven
Electronic publication date: 2023 Aug 10
Publication date: 2023
Volume: 9
Electronic Location ID: e1476
Received 2022 Nov 2; Accepted 2023 Jun 12
Copyright: ©2023 Rashid et al.
Copyright year: 2023
Copyright holder: Rashid et al.
License: This is an open access article distributed under the terms of the Creative Commons Attribution License, which permits unrestricted use, distribution, reproduction and adaptation in any medium and for any purpose provided that it is properly attributed. For attribution, the original author(s), title, publication source (PeerJ Computer Science) and either DOI or URL of the article must be cited.
License URL: https://creativecommons.org/licenses/by/4.0/

Keywords: Nonlinear exploration, In-depth browsing, Usability, Visualization, Search user interfaces, Web search

Funding: Prince Sultan University The authors received support from Prince Sultan University for the Article Processing Charges (APC) of this publication. The funders had no role in study design, data collection and analysis, decision to publish, or preparation of the manuscript.

==============================
The existing image search engines allow web users to explore images from the grids. The traditional interaction is linear and lookup-based. Notably, scanning web search results is horizontal-vertical and cannot support in-depth browsing. This research emphasizes the significance of a multidimensional exploration scheme over traditional grid layouts in visually exploring web image search results. This research aims to antecedent the implications of visualization and related in-depth browsing via a multidimensional cubic graph representation over a search engine result page (SERP). Furthermore, this research uncovers usability issues in the traditional grid and 3-dimensional web image search space. We provide multidimensional cubic visualization and nonlinear in-depth browsing of web image search results. The proposed approach employs textual annotations and descriptions to represent results in cubic graphs that further support in-depth browsing via a search user interface (SUI) design. It allows nonlinear navigation in web image search results and enables exploration, browsing, visualization, previewing/viewing, and accessing images in a nonlinear, interactive, and usable way. The usability tests and detailed statistical significance analysis confirm the efficacy of cubic presentation over grid layouts. The investigation reveals improvement in overall user satisfaction, screen design, information & terminology, and system capability in exploring web image search results.

Introduction

Nowadays, the production and sharing of web images are exponential. The proceeding years have witnessed tremendous growth in the production of web images (Zhang et al., 2019). Over the web, the volume of images is vast and plays a significant role in satisfying users’ information needs (Göker et al., 2017). The users also explore web images to meet their visual information needs (Kofler, Larson & Hanjalic, 2016). The users generally express visual information needs by giving short and ambiguous textual query terms (André et al., 2009). Consequently, web search engines exploit textual information modality associated with images (as accompanying text, tags, descriptions, meta information, etc.) in exploration. The intention is to provide a variety of exploration layouts to the users via a particular search engine result page (SERP) (Vo et al., 2019; Yuan & Gao, 2019; Klostermann et al., 2018).

The adaptation in the elementary interfaces, which provides visual exploration, still needs improvements (Hoque, Hoeber & Gong, 2013; Keisler et al., 2019); e.g., Google Image Search (https://images.google.com), Bing (https://www.bing.com), Flickr (https://www.flickr.com) etc., employs traditional grid layouts in exploration. On the contrary, advanced exploration and reach-ability possibilities discussed in literature recommended representing results in complex and nonlinear data models (i.e., trees, graphs, hierarchies, cluster, classes, etc.) and associated exploratory interfaces (Rashid et al., 2016; Saddal, Rashid & Khattak, 2019; Xie et al., 2018). The subsequent exploration possibilities are nonlinear, where users can interact with web search results organized in complex graphical layouts (Yan et al., 2017; Rayar et al., 2018). The researcher revealed the efficacy of user interaction with SERPs by employing nonlinear interactive ways using various exploration interfaces and nonlinear data models, particularly in multimedia information retrieval and aggregated search (Rashid & Bhatti, 2015; Rashid et al., 2016; Rashid & Bhatti, 2017; Rashid, Saleem & Ahmed, 2021).

The readiness of advanced exploration options to integrate into image exploration requires further investigation. Web image search results retrieved against textual query terms could be better organized in cubic graphs by exploiting the relationships associated with textual annotations. The mentioned representation could be further exploited to provide nonlinear exploration options to users. In addition, cubic graphs could be visualized to provide navigation and in-depth browsing of web image results. Using cubic graphs to represent results in result spaces leads to the introduction of novel exploration tools to interact with results retrieved against textual query terms in web image searches.

In this research mainly, we retrieve the web images and associated textual information like titles, descriptions, tags, URIs, etc., from Flickr image sources against textual query terms. We ranked the results by considering their textual similarity and query terms using a normalized vector space model. In the ranking, we exploited the textual information annotated with the web image results to represent further the ranked results in a particular cubic graph layout, where nodes are web image results and edges are textual similarity relationships. Finally, we visualized web image search results retrieved against textual query terms in the form of an interactive and browsable cubic layout. We also compared web image exploration via cubic layout with the traditional grid layout provided by Flickr via various usability tests. In the following, we highlight the limitations of the existing approach. We also discuss the research’s motivational factors, research questions, and contribution.

Limitations

The utilization of images is experiencing exponential growth in the present era (Bouchakwa, Ayadi & Amous, 2020). The web image search results representation over SERPs is primarily linear, where results are ranked in lists by considering the textual similarity of query and retrieved web image results (Al-Jubouri, 2019). The SERPs usually present the linearly organized web image search results via primitive grid layouts, typically 2-dimensional layouts (Alzubi, Amira & Ramzan, 2015). The in-depth browsing of image results is not possible (Kopalle et al., 2022). Recently research has been focusing on image search result organization as a viable approach to enhance the quality of image retrieval on the web (Tekli, 2022). The traditional organization of web image search results in 2-dimensional grids allows horizontal-vertical scanning of the ranked image results. However, the results at lower-ranking positions could be more challenging to reach and navigate (Rashid, Saleem & Ahmed, 2021). The complexity associated with advanced approaches makes them cumbersome to instantiate over the real dataset of web image search results since the graphs are dense (Rashid, Saleem & Ahmed, 2021). Navigating the desired results in the dense graph could be less convenient. The linear adaptation of SERPs permeates globally to provide efficacy in exploring web image search results. However, the focus is not to give the in-depth browsing of web image search results organized in traditional grid layouts.

Motivations

The in-depth browsing that allows the users to navigate non-linearly within the web image search results presented in standard grid layouts inspired us to conduct this research. We want to organize the web image search results retrieved from web sources in cubic graphs and enable their exploration via a particular SUI design; so the users can perform in-depth browsing along with traditional horizontal-vertical scanning in an interactive, integrated, and usable way. We are particularly interested in reaching non-linearly web image search results at lower-ranking positions. To do this, in this research, we propose the I-Cubid approach. In particular, by the I-Cubid approach, it would be possible to: (i) retrieve, visualize, and in-depth browse the retrieved web image search results from the online web source, i.e., Flickr, and (ii) explore a cubic graph space of web image search results via various components of a comprehensive SUI design.

Research questions

The linear ordering of web image search results presented in traditional grid layouts may not support in-depth browsing, navigation within the search results, and reachability. This research focuses on representing the web image search results in cubic graphs and enabling their exploration via a particular SUI design. We formalized the following research questions:

• RQ1: How can a cubic graph and associated SUI design be exploited mainly in the ranked representation of web image search results to support visualization and in-depth browsing activities?

• RQ2: Does a cubic graph result space instantiated over web image search results provide visualization, in-depth browsing, and address reachability issues in an integrated way?

• RQ3: Is the interaction with web image search results in visualization and in-depth browsing provided via the I-Cubid approach usable and satisfying the users’ information needs?

• RQ4: Do users prefer the interaction as visualization and in-depth browsing via cubes in the I-Cubid approach over the traditional 2-D grid-based layout in web image results exploration?

The readiness of advanced exploration options to integrate into exploration requires further investigation. The research questions discussed in this section addressed the particular research directions. Notable, RQ1 addresses the formation of a cubic graph and SUI design; RQ2 describes the usage of instantiated results space and interaction with it by the users; RQ3 reveals the usability and its associated dimensions; RQ4 provides the comparison of usability related to the proposed SUI design and standard grid layouts.

Contributions

Our approach used cubic graph representation to visualize web image search results. The users can browse the web image results in a usable way, view details associated with image results, and access images and their associated information from their sources in an integrated manner. The users can navigate horizontally-vertically and across the planes (in-depth) dynamically to explore the web image search results in a usable way. Our approach enables the nonlinear exploration possibilities to interact with the cubic graph of image results. The approach is further compared with traditional grid-based layouts. The cubic visualization gives satisfactory mechanisms to explore the web image search results. The exploration of results via cubes passed various usability tests and was comparatively more convenient than traditional grid layouts in different usability dimensions.

Following is the organization of this paper. In Section 2, we discuss the literature review. In Section 3, we describe the I-Cubid: approach. Section 4 discusses the research methods employed to validate the I-Cubid approach. In Section 5, we analyze the experimental results. Finally, in Section 6, we conclude our discussion and highlight future research.

Literature Review

Background

The retrieval techniques provide access to web image repositories by exploiting the textual and visual information modalities associated with them (Unar et al., 2019; Zha et al., 2020). The former as keywords with web images as textual descriptions, annotations, tags, meta information, etc., and later as low-level visual features extracted from them (Datta et al., 2017; Saddal, Rashid & Khattak, 2019). However, existing retrieval techniques usually employ textual modalities to explore the web image search results. The search paradigm consequently satisfies information needs by exploring image results retrieved by considering queries, search intents, and relevance feedback (Wu et al., 2019a). The satisfaction of information needs via the visual search paradigms is linked to information needs expression and browsing of the retrieved results via SERPs (Zhang et al., 2018; Wu et al., 2019b). Over the years, linear and nonlinear exploration layouts emerged as separate paradigms. In the following, we discuss linear, nonlinear, and state-of-the-art image results exploration research.

Linear exploration

Traditionally web image search results interaction usually employs linear interaction layouts to provide exploration, where users can give textual query terms (usually via bi-grams and tri-grams) or visual words (image examples) (Jégou, Douze & Schmid, 2010). Consequently, interaction with search results via grid layouts is possible, which generally supports exploration via lookup-based activities (Vrochidis et al., 2019). In this case, the support for the exploratory activities could be minimal. The only option is to interact with the image results via 2-dimensional scanning of grid-based layouts, where the users can traverse results horizontally-vertically (Xie, 2019; Xie et al., 2019). The grids may consider the similar relationships between the query and textual information associated with web image search results. Notably, most grid layouts do not consider the similarity and semantic relationships in the image results. However, the grids usually represent the ranking order, which may decrease from top-bottom and left–right. The interaction options are restricted due to the limitations associated with them. Figure 1 elaborates on exploring web image search results via a linear interaction paradigm. As it emerges from Fig. 1 that the exploration is possible via grid layouts, image thumbnails, facets, color schemes, and cluster tags in web image search engines. However, they provide 2-D grid plane exploration. The plane represents images as a single-layer 2-D grid consisting of rows and columns. Each plane contains a set of highly similar images. Further in-depth browsing of a selected image plane is not possible.

Nonlinear exploration

The nonlinear exploration approaches usually organize the web image search results in nonlinear data models, e.g., graphs, trees, multi-lists, knowledge networks, self-organizing maps, etc., by exploiting similarity and semantic relationships. The nonlinear representation is further exploited in exploring web image search results. The focus is to provide the visualization of results that enhance the users’ exploration activities to seek the relevant images. The nonlinear exploration facilitates the exploratory search tasks (Vrochidis et al., 2019). The SUI is usually based on the underlying nonlinear organization of the web image search results. The user can navigate results non-linearly and perform exploration activities, including navigation, in-depth browsing, visualization, and access to the desired results (Rashid, Saleem & Ahmed, 2021). However, exploring web image search results via dense and nonlinear data models is challenging via the interactive exploration tools (Yan et al., 2017). The mechanism of the advanced interactions exploits the dynamic grid layouts, clusters, heat-maps, etc., in the nonlinear exploration over Picsbuffet (Jung et al., 2021), Google Image Swirl (Jing et al., 2012), t-SNE algorithm (Wattenberg, Viégas & Johnson, 2016), and interactive graph (Rashid, Saleem & Ahmed, 2021).

Figure 1 Linear exploration of image results via grid layouts, image thumbnails, facets, and tagged clusters over (A) Flickr, (B) Google Image search, (C) Getty, and (D) Pexels.

State-of-the-art

Grid layouts

AIRS (Doloc-Mihu et al., 2005), VisionNext (Pu, 2005), RETIN (Gony et al., 2007), PHOTO-BASED QA (Yeh, Lee & Darrell, 2008), Uinteract (Liu et al., 2009), MQSearch (Luo et al., 2008), img(Anaktisi) (Zagoris et al., 2009), ViewFinder, VOIR (Torres & Reis, 2008), PicSOM (Laaksonen et al., 2000; Albertson, 2010), PatMedia (Vrochidis et al., 2010), and ImageHunter (Tronci et al., 2013) are popular web image search engines, discussed in the literature, provide grid-based interaction with results. Particularly, AIRS and RETIN modify the image results by taking relevance feedback from the users as relevant and irrelevant; VisionNext provides the access to the textual information associated with image results; PHOTO-BASED QA enabled the question-answer based interaction with image results; VOIR framework enabled the retrieval refinements by exploiting the user highlighted image results; Uinteract categorized results as negative and positive examples and exploits them in results set refinements; MQSearch visualized the image results retrieved via multiple user queries; PicSOM exploited user relevance feedback in similarity calculations and also provides the access to previously retrieved image results; img (Anaktisi) retrieved results can automatically readjusted in searching through auto relevance feedback mechanisms; ViewFinder refined the image results by selecting various image features and accompanying textual details; PatMedia provided multiple separate displays for figures, patents, and pages to interact with the search results; ImageHunter enabled dragging and dropping of relevant images in a separate in further relevance feedback calculations. Similarly, Google Image search, Bing Image search, Ask Image search (http: //www.ask.com), Yahoo Image search (https://images.search.yahoo.com/), Picsearch (http://www. picsearch.com), Exalaed Image search (https://www.exalead.com), Terra Galleria (http://www.terragalleria.com/), Flickr, Photobucket (http://photobucket.com/), CorbisImages (http://www.corbisimages.com/), and Getty Images (http://www.gettyimages.com/) are online image search engines available over the web and provides the visualization of image search result thumbnails in grid-based layouts.

The recent trend is focusing on the human interaction aspects during image exploration. For instance, Dziubak & Bunt (2018) provides interactive grid-based (SUI) that automatically captures all explored images and annotates them with the search trails. Similarly, users are becoming interested in multimodal interaction (Rashid, 2017). Various approaches have been devised to connect an image and video content based on similarity (Rashid, 2017; Rashid, Saleem & Ahmed, 2021; Khan et al., 2021). A few of the state-of-the-art image search engines, such as Pexels (http://pexels.com/), Morguefile (https://morguefile.com/), Vecteezy (https://www.vecteezy.com/), etc., are incorporating video artifacts in image exploration. However, despite technical advancements, most of these approaches still provide grid-based interaction. Therefore, a need exists to investigate multidimensional approaches to explore images and multimedia data.

Graph layouts

ImageSEE (Gaikwad & Hoeber, 2019), FACETS (Pienta et al., 2017), Apic (Deeswe & Kosala, 2015), Swirl (Jing et al., 2012), DynamicMaps (Kleiman et al., 2015), PRiSMA (Tolchinsky, Chiarandini & Jaimes, 2012), etc., are examples of web image search engines discussed in literature providing the nonlinear interaction with results. ImageSEE enabled the image results exploration in the pre-focus and post-focus stages. The former extracts and visualize the augmented image results (via hashtags and timestamps) in specialized cascaded grids; focus formulation allows pan and zoom action on visualized image results to extract desired information while later enabling the seeking of the filtered workspace as a grid of image results, in an interactive and integrated way. FACETS progressively visualized image results as graph nodes in distinct facets. The tool enables the orderly exploration of relevant and their neighbor nodes organized in facets non-linearly. Various colors in visualized graph depict users’ exploration activities and exciting topics. Apic provides a 3-dimensional tree visualization interface to explore Bing image search results. Exploiting the textual relationships represents retrieved and relevant image results as tree nodes. The users can access retrieved results and expand the exploration activities by accessing retrieved relevant results from an interactive 3-dimensional tree view. Swirl provided large-scale browsing of image results by collecting the top 1,000 image results of 400K queries from Google. It organizes image results in hierarchical clusters by exploiting visual features and enables the exploration of image results by browsing hierarchical clusters. DynamicMaps allows browsing large sets of image objects by considering their feature-based similarity. The tool dynamically places the image objects on a 2D grid next to their nearest proximal neighbors in a high-dimensional feature space. The 2-D canvas provides a continuous browsing and navigation experience via an infinite 2D grid-based layout. PRiSMA promotes image results exploration via multiple linear strips. The users can explore various horizontal sliding strips to interact simultaneously via different perspectives (locations, topics, features, etc.) associated with collections. The users can easily create, tailor, merge, and remove strips to allow various associated dynamic and orderly exploration activities.

I-Cubid interaction

Approach

The proposed I-Cubid approach exploits a particular nonlinear ranked representation to explore the web image search results retrieved from Flickr in real time against the user-issued queries. While the proposed approach can be adapted to any image dataset, we chose Flickr as a baseline for realistic results comparability. It provides exploration via a particular 3-D cubic structure and interaction with the web image search results in various ways. In the I-Cubid approach, users can express their visual information needs by introducing simple textual query terms (usually bi-grams and tri-grams) and interact with web image search results via multiple integrated layouts. However, the approach primarily focuses the nonlinear navigation as in-depth browsing within the retrieved web image search results. The objective is to enable the nonlinear exploration of web image search results in a usable way. Figure 2 illustrates the I-Cubid approach.

The I-Cubid approach enables the visual information need expression via an interactive SUI. The users can formulate the textual queries for visual search by giving the multiple keywords connected via Boolean (AND, OR, NOT) operators (Fig. 2A). The keywords are initially transferred to a meta-search engine, which pre-processes textual query terms (performs necessary text indexing operations, i.e., stop-word removal and stemming) and dispatches the user query terms to a selected web image search engine (Fig. 2B). The results are retrieved from the Flickr database via indexes of web search engines since Flickr supports the textual indexing and retrieval of images (Fig. 2C). The results retrieved against the textual query terms from Flickr are initially stored in a web image results space, where the association within the results is unknown (Fig. 2D). Initially, our approach ranks web image search results in a 2-D grid layout (Fig. 2E). The I-Cubid approach employs a particular algorithm to compute inter-image similarity ranking to represent the results in a 3-D cubic data model (Fig. 2F).

Figure 2 Schematic representation of I-Cubid approach containing the (A) Search User Interface, (B) Retrieval mechanism, (C) Web Source, (D) Un-ranked image retrieval space, (E) image grid retrieval space, (F) inter-image similarity ranking and cube modeling algorithm , and (G) Cubic image retrieval space.

The I-Cubid approach represents the web image search results in separate 2-D planes. Finally, it assembles them in a 3-D cubic representation. It is further utilized as a results space for the visualization and in-depth browsing of Flickr images retrieved against textural query terms (Fig. 2G). In the I-cupid approach, the users can explore the web image search results via a navigable and dynamic 3-D cubic visualization scheme. The users can initiate image search results-seeking activities by traversing (top-down and left–right) 2-D grids and in-depth browsing of 3-D cube planes of web image search results. Initially, the retrieved web image search results are ranked in a grid-based layout by considering the textual similarity of the query and textual information associated with the retrieved image results since our approach also retrieves the textual information associated with web image search results (Fig. 2C).

The I-Cubid approach employs the inter-image textual similarity ranking algorithm and cube modeling algorithms to transform the web image search results represented via 2-D grid-based layouts into multiple 2-D planes and 3-D cubic representations. The I-Cubid approach visualizes 3-D cubic representations to the users via an interactive SUI design. I-Cubid mainly provides the visualization and in-depth browsing of Flickr images by employing a 3-D cube and SUI design; the former represents the web image search results retrieved from the Flickr database against textual query terms and later visualizes them in a variety of integrated and usable ways. We will discuss the 3-D cube formalization, SUI design, and usage scenario in the following.

Cubic graph formation

Let I be the web image search result space retrieved against a user-based query formulation. The I = i1, i2, …, in shows there is n number of retrieved web images comprising textual metadata D=.... The I is visualized in grid structure G and cubic structure making it a cubic graph data model ICG to enable exploration activities (Fig. 3). Formally a graph model is represented as an acyclic, labeled graph G = (V, E) and ICG = G therefore, ICG = (V, E), where V is image nodes and E is the non-negative weighted edges where E=e1,e2,…,en and V=v1,v2,…,vm.

Figure 3 Cubical graph formation (A) cube, (B) multiple cubic planes, (C) outer corner image nodes of p1⋀pn, (D) outer image nodes except corners of p1⋀pn, (E) inner image nodes of p1⋀pn, (F) outer corner image nodes of p2, …, pn−1, (G) outer image nodes except corners of p2, …, pn−1, (H) inner image nodes p2, …, pn−1.

The E counts the similarity relationships between V as the weights. It means edges display the similarity between images. The ICG have multiple planes P which forms a cube shape where P=p1,p2,…,pn, shows n number of planes depending upon the I (Figs. 3A, 3B). The maximum plane Pmax should Pmax ≤ n and the maximum number of planes visualized Pvmax at a time Pvmax = 5. Each P in ICG have its neighbor image nodes Ine=ine1,ine2,…,inen where Ine ∈ I and Ine ∈ P⋀ P ∈ ICG∴Ine ∈ ICG.

The arrangement of I depends upon E in ICG; consequently, Ine are generated. The position of Ine in P also depends upon E in ICG and ∀in∃en. The degree of an image node deg(n) is the number of adjacent edges presenting the Ine and ∀en the degree range is 3 ≤ deg(n) ≤ 6 to form ICG. Keeping in view, the following case properties hold:

• In the case outer corner image nodes of p1⋀pn, ∀en ∈ ICG the degni=3→Ine=3 (Fig. 3C).

• In the case of outer image nodes except corners of p1⋀pn, ∀en ∈ ICG the degni=4→Ine=4 (Fig. 3D).

• In the case of inner image nodes of p1⋀pn, ∀en ∈ ICG the degni=5→Ine=5 (Fig. 3E).

• In the case of outer corner image nodes of p2, …, pn−1, ∀en ∈ ICG the degni=4→Ine=4 (Fig. 3F).

• In the case of outer image nodes except corners of p2, …, pn−1, ∀en ∈ ICG the degni=5→Ine=5 (Fig. 3G).

• In the case of inner image nodes p2, …, pn−1:∀en ∈ ICG the degni=6→Ine=6 (Fig. 3H).

The Ine are created on the based of degni according to the properties to generate P forming ICG. The E is created if the edge weight W exceeds the threshold γ. The details of thresholds, number of planes, and similarity measures are discussed in the other sections.

3-D cubic in-depth browsing and navigation

The I-Cubid graph data model initially represents web image search results into 2-D planes and further generates 3-D cubic representation. Moreover, it provides a web image search results space visualization and in-depth browsing of Flick images. The users initiate exploration activities via navigable and dynamic 3-D cubic web image search result arrangement. Users commence their image-seeking activities by in-depth browsing of 3-D cube planes. The web image search results-based 3-D cubic representation comprises multiple planes, and the browsing pattern involves the image’s traversing and plane’s switching (Fig. 4). The image position can be different in the same plane and can navigate to other planes to browse deeply in web image search results. Figure 4 navigation path shows that browsing initially begins with the first plane p1 center position, then shifts upwards and moves to the second plane p2 similar to the other planes till the last plane pn.

SUI design

This section elaborates on an SUI design for cubic visualization and exploration. The SUI design consists of multiple panels providing interaction with Flickr images. The main objective of SUI design is the cubic visualization of the Flickr images to provide in-depth browsing and navigation in the cube exploiting the image relationships. Figure 5 depicts an SUI design comprising multiple panels. The details of the SUI design’s panels are as follows.

Query panel (QP): The QP allows users to fulfill their information needs by formulating textual queries (Fig. 5A). The textual queries enable visual search by giving multiple keywords via Boolean keywords (AND, OR, NOT). The simple textual query terms are usually bi-grams or tri-grams connected via Boolean keywords fulfilling the users’ exploration needs. These Boolean keywords activate the Boolean operations between the query keywords to filter the image retrieval result space. In this case, the user issues the query “cat” in the QP, which is dispatched to the Flickr database in real time.

Grid view panel (GVP): The GVP shows the query-based image search results from Flickr in a standard 2-D grid layout (Fig. 5B). The clickable image thumbnails maintain their original aspect ratio leading toward textual information. Initially, the images retrieved against the “cat” query in the 2-D grid utilize the Flickr ranking. The checkbox on the GVP enables Jaccard-based ranking, computed by relevancy between query keywords and image meta-tags. It shows the number of relevant images found on Flickr as well. The GVP allows to transverse the image horizontally-vertically in a plane and navigate to the cubic browsing panel via the cube icon. During the exploration, the user selects the most appealing “cat” image in the initial to further explore the related images (Fig. 4 (P1)).

Figure 4 Cubical graph navigational path showing the cubical plane navigational support from initiation (P1) till satisfaction of information need (Pn).

Figure 5 I-Cubid SUI design (A) query panel, (B) grid view panel, (C) grid image view panel, (D) cubic browsing panel, (E) cubic image view panel, (F) cubic navigation panel.

Grid image view panel (GIVP): The GIVP displays an image with its textual metadata (Fig. 5C). After a user selects an interesting “cat” image, it presents a chosen image thumbnail in the original aspect ratio. The meta-information provides views, favorites, and comments on Flickr. It elaborates on the selected image format (Fig. 4 (P1)) and the date and time of the image taken. The GIVP also provides an option to download the image.

Cubic browsing panel (CBP): The CBP visualizes the images in a 3-D cube layout (Fig. 5D). The 3-D image cube comprises multiple planes (Fig. 4 (P1 ... Pn)), and each plane contains 25 distinct “cat” images related to the clicked “cat” image. The image thumbnails having a fixed aspect ratio are presented in each plane. The CBP allows browsing within the multiple cubic planes to enable in-depth exploration. Unlike traditional grid exploration, users can view any cubic plane via the ‘switch plane’ option to further explore the related images. Initially, the cube images use Flickr-based ranking, and the checkbox ranks the cube images via Jaccard similarity. The selection of image display in planes is based on Jaccard similarity ranking and descending order sorting. We chose Jaccard similarity instead of embeddings, such as Glov (Pennington, Socher & Manning, 2014; Mikolov et al., 2013), due to its least computationally requirements and to avoid sparsity problems connected to a vector-based representation (Rashid & Bhatti, 2017). Each cube shows a fixed number of planes, i.e., five cubic planes of the selected “cat” image, at a time. In cases where the number of images exceeds the five planes, users can navigate to ‘next’ to view more planes of the same cube.

Cubic image view panel (CIVP): The CIVP exhibits the selected image thumbnails in the original aspect ratio (Fig. 5E). It shows the image’s metadata, including the number of views, favorites, comments, and image creation details, i.e., time and date. It also specifies the image format and allows to download of that particular image. The CIVP enables the view of the neighboring “cat” images related to the selected “cat” image (Fig. 3).

Cubic navigation panel (CNP): The CNP enables the navigation in the neighboring images of select “cat” image (Fig. 5F). The neighbor image includes the images from the same plane, e.g., in the case of the second plane as shown in (Fig. 4 (P2)), the neighbor includes left, right, top, bottom, and the images from deeper planes, e.g., front and back images (Fig. 4 (P1 and P3)). Hence, the neighbors are computed based on the arrangement in the 3-D cube layout.

I-Cubid interaction mechanism with SUI design

The I-Cubid interaction enables users to express information needs, instantiate exploration, 2-D nonlinear navigation, and 3-D nonlinear navigation via SUI panels (Fig. 6).

Figure 6 I-Cubid interaction mechanism, outlining (A) query dispatching, (B) retrieval of search results, from (c) Flickr online source, (D) instantiation of the cubic data model, displaying the results on the (E) traditional grid view panel, and (F) cubic image view panel, further navigating to (G) Flickr dataset, via (H) cubic navigation panel, on (I) cubic browsing panel, and finally ranking (J) cubic, and (K) grid results.

Expressing information needs: The users express their information needs via keywords-based queries. The QP allows keyword submission to initiate the exploration activities. The QP is directly connected to other SUI panels (Fig. 6A).

Instantiating exploration: The users can perform traditional 2-D grid-based browsing in GVP. The image result space depends upon dual rankings (Flickr-based and I-Cubid based), enabling exploration (Figs. 6B, 6K). The 2-D grid-based exploration leads towards associative panels, i.e., GIVP and CBP, to enable 2-D and 3-D nonlinear navigation (Figs. 6L, 6D). The objective is to give users an overview of the Flickr-based retrieved image result space-providing exploration activities.

2-D nonlinear navigation: The users can navigate non-linearly in the traditional 2-D grid via GVP. It allows one to view selective image thumbnails and browse within the GIVP (Fig. 6L). It will enable the access of web image search results non-linearly and horizontally-vertically traversal. The associative details are displayed on GIVP, which further navigates to the actual web image source on Flickr (Fig. 6E). The web image search result space arrangement depends upon dual rankings (Flickr-based and I-Cubid-based), enabling nonlinear navigation. The 2-D navigation of GVP and GIVP leads towards CBP enabling 3-D nonlinear navigation.

3-D nonlinear navigation: The CBP provides 3-D navigation and in-depth browsing. The CBP shows the image result space in a 3-D cube structure allowing users to navigate within the cube and its multiple planes. The cube provides also provides a dual ranking and allows one to navigate back to GVP (Figs. 6J, 6D). The CBP is also connected with other cube panels like CIVP and CNP (Figs. 6F, 6H). The users can browse associative image details on CIVP, which further navigates to the actual Flickr source (Fig. 6G). The cube structure consists of multiple planes that allow 3-D navigation via CNP (Fig. 6H). In CNP, each cube plane is connected with other planes showing retrieved images to activate in-depth browsing, and each image highlights its neighboring images (left, right, top, bottom, front, and back). The users can even navigate back to the CBP, enabling in-depth browsing and exploration (Fig. 6I).

Implementation details

We implemented I-Cubid as a web application. The backend was developed using the PHP scripting language suited to web development. The backend contained the logic to retrieve the image results from the Flickr publicly available API in real-time against user queries and process it into the nonlinear cubic graph. The front end was developed using HyperText Markup Language (HTML), Cascading Style Sheet (CSS), and JavaScript to provide the interaction with the search user interface. We used BootStrap (https://getbootstrap.com/) and Vue.js (https://vuejs.org/) libraries to make the search user interface responsive across various screen sizes. The I-Cubid project was managed on the Visual Studio (https://visualstudio.microsoft.com/) integrated development environment. The website was hosted on the Apache XAMPP (https://www.apachefriends.org/) server. From the performance perspective, the proposed tool, on average, took 5 s from query dispatching to search results retrieval on the search user interface. The further hardware details are discussed in section 4.1.

Evaluation

The I-Cubid interaction enables the nonlinear exploration of web image search results retrieved from online sources in a usable way. On the contrary, the existing web image exploration approaches present the results in ranked grid layouts and evaluate them in terms of standard precision and recall measures. The current approaches need more focus on evaluating the user exploration experience. The recent literature recommends assessing the users’ exploration experiences via state-of-the-art usability tests (Rashid, 2017; Rashid, Saleem & Ahmed, 2021). In this research, we also employed the standard usability tests to evaluate (RQ3) and compare the I-Cubid approach (RQ4). In the following, we discuss the data set, instantiation, and evaluation methodology adopted to evaluate the I-Cubid interaction.

Apparatus & dataset

The I-Cubid tool was installed on a lab computer with Windows 10 Pro operating system, 10 GB RAM, and a 3 GHz dual-core processor. Access to the computer was provided with a wired mouse and keyboard. The screen recording software was also installed on the computer to record users’ interactions with the tool. In this study, we employed a real dataset of Flickr images retrieved via simple textual query terms to instantiate the I-Cubid interaction.

Users

We recruited 18 users for the usability evaluation of the I-Cubid tool. According to existing literature, a minimum of five users are required to uncover 80% of the usability issues, and 12 users are adequate for obtaining statistically significant results (Alroobaea & Mayhew, 2014). To make the evaluation process more rigorous, we recruited users from diverse backgrounds (housewives, students, and professionals). Among them, 10 were males, and eight were females. The users’ minimum, maximum, and mean ages were 21, 45, and 28 years, respectively. The users’ complete details are provided in Table A1.

Study design & instruments

We employed within-subjects usability evaluation. The users were asked to evaluate both systems; baseline and I-Cubid. The baseline system provided a grid-based image presentation, and I-Cubid provided a 3D cube-based image presentation. The users were initially given a demographic survey and brief instructions for evaluating and simulating the tools. This introductory session consisted of 5 to 10 min. Afterward, the users were asked to choose a topic of interest from four available issues. Finally, the Questionnaire for User Interface Satisfaction (QUIS) (Chin, Diehl & Norman, 1988) and System Usability Scale (SUS) (Brooke, 1996) instruments were deployed to measure the system usability and user interface satisfaction on both systems. The affiliated university board duly approved this study design (Ethical Application Ref: UR \BSCS \Projects \0039). Furthermore, verbal consent from the participants was taken before the conduct of human studies (Table A2).

Topics & Tasks

We designed four generic topics; “Gold Fish in Aquariums”, “Bullet Trains in Japan”, “Fighter Aircraft in Pakistan”, and “Indian Tiger”. These exploration topics were designed for usability evaluation. Each topic consisted of a guided task with complete procedural instructions provided to the user and an unguided task without guided instructions. The guided task aimed to increase users’ familiarity with the tool to reduce the learning effect. The tasks were designed to evaluate the grid view panel and cubic panel. Tasks were performed by users using the topic of their own choice. The topics and tasks details are displayed in Table A3.

Results

System usability

The SUS administered to the users shows an overall average score of 80.8 on the grid-based image presentation. According to the ranking yardstick presented in Lewis & Sauro (2018), the grid presentation achieves a grade ‘A’ in the usability test and falls into the top 90–95 percentile of the best usable tool. On the I-Cubid, the system usability administered to the users reported an overall average score of 78.2. According to the ranking yardstick presented in Lewis & Sauro (2018), it achieves a grade of ‘B+’ and falls into the top 80–84% of the usable performing tools. The overall SUS results for the cube and grid-based system, including T-Test, distribution frequency (df), mean differences, standard deviation, and error mean, are displayed in Table 1.

Table 1 Grid-based and cube-based statistics testing analysis for the SUS questionnaire.

Statistical test	Q/System	Q1	Q2	Q3	Q4	Q5	Q6	Q7	Q8	Q9	Q10	
t		Grid	17.23	8.17	16.03	7.08	17.06	9.90	12.00	11.66	30.88	7.79	
	Cube	20.28	6.54	14.75	7.20	16.95	7.81	24.14	9.07	26.75	8.74	
df		Grid	17.00	17.00	16.00	17.00	17.00	17.00	17.00	17.00	17.00	17.00	
	Cube	17.00	16.00	17.00	17.00	17.00	17.00	17.00	17.00	17.00	17.00	
Mean difference		Grid	4.06	1.72	4.29	1.83	4.11	1.83	4.00	1.33	4.50	1.67	
	Cube	4.28	2.06	4.11	2.22	4.17	1.94	4.28	1.83	4.44	2.06	
95% confidence interval of the difference	Lower	Grid	3.56	1.28	3.73	1.29	3.60	1.44	3.30	1.09	4.19	1.22	
Cube	3.83	1.39	3.52	1.57	3.65	1.42	3.90	1.41	4.09	1.56	
Upper	Grid	4.55	2.17	4.86	2.38	4.62	2.22	4.70	1.57	4.81	2.12	
Cube	4.72	2.73	4.70	2.87	4.69	2.47	4.65	2.26	4.79	2.55	
Mean		Grid	4.06	1.72	4.29	1.83	4.11	1.83	4.00	1.33	4.50	1.67	
	Cube	4.28	2.06	4.11	2.22	4.17	1.94	4.28	1.83	4.44	2.06	
Std. deviation		Grid	1.00	0.89	1.10	1.10	1.02	0.79	1.41	0.49	0.62	0.91	
	Cube	0.89	1.30	1.18	1.31	1.04	1.06	0.75	0.86	0.70	1.00	
Std. error mean		Grid	0.24	0.21	0.27	0.26	0.24	0.19	0.33	0.11	0.15	0.21	
	Cube	0.21	0.31	0.28	0.31	0.25	0.25	0.18	0.20	0.17	0.24	
Significance/P-Value (α = 0.05)	Grid/Cube	0.00	

User interface satisfaction

The QUIS scale measures five categorical aspects via several questions about the administered tool. These categories measure (1) quality of interface design (such as readability, organization, and visual aids, etc.), (2) terminology and system information (including ease of use and user-friendliness), (3) user learnability, (4) quality of developed tool in terms of speed, accuracy, and usefulness, and finally (5) overall perceived reaction to the tool. This scale includes a total of 27 questions to measure all the categorical aspects of the tool.

The users on the grid-based interface reported an overall average of 83% on the QUIS. Therefore, grid presentation is perceived highly satisfactory among the users in terms of the usability and interface satisfaction. For the user interface satisfaction, the cube-based images presentation overall achieves an average of 83.2%. The overall QUIS results for the cude and grib-based system are displayed in Table A5 and Table A7, respectively.

UI quality.

Holistically, both image browsing paradigms (grid and cube) were praised similarly by the users (82% and 83.33%, respectively), according to the obtained scores provided in Table 2. The results were further statistically analyzed to find the significance of the results using a t-test. According to the results shown in Table 2, no statistical discrepancies were found, and therefore, the results achieved statistical significance when tested on the confidence interval of 95%.

Table 2 Grid-based and cube-based statistics testing analysis for the QUIS (Screen) questionnaire.

Statistical test for UI	Q/System	Q1	Q2	Q3	Q4	
t		Grid	15.94	14.21	13.83	16.00	
	Cube	15.16	21.32	16.11	19.18	
df		Grid	16.00	16.00	16.00	16.00	
	Cube	16.00	17.00	17.00	17.00	
Mean difference		Grid	7.71	6.94	7.41	7.53	
	Cube	7.59	7.56	7.61	7.72	
95% confidence interval of the difference	Lower	Grid	7.59	7.56	7.61	7.72	
Cube	6.68	5.91	6.28	6.53	
Upper	Grid	8.73	7.98	8.55	8.53	
Cube	6.53	6.81	6.61	6.87	
Mean		Grid	7.71	6.94	7.41	7.53	
	Cube	7.59	7.56	7.61	7.72	
Std. deviation		Grid	1.99	2.01	2.21	1.94	
	Cube	2.06	1.50	2.00	1.71	
Std. error mean		Grid	0.48	0.49	0.54	0.47	
	Cube	0.50	0.35	0.47	0.40	
Significance/P-Value (α = 0.05)	Grid/Cube	0.00	

Ease-of-Use.

Regarding ease of use, the cube-based approach achieved the result of 84.56%, and the grid-based approach achieved 80.78% overall scores. Therefore, the cube approach was found to be better than the grid. The results were further statistically analyzed to find the significance of the results using a t-test. According to the results provided in Table 3, no statistical discrepancies were found; therefore, the results achieved statistical significance when tested on the confidence interval of 95%.

Table 3 Grid-based and cube-based statistical analysis for the QUIS (Info.) questionnaire.

Statistical test for Info.	Q/System	Q1	Q2	Q3	Q4	Q5	Q6	
t		Grid	15.59	16.48	16.14	14.24	11.08	7.97	
	Cube	18.07	22.64	15.15	16.27	14.95	8.11	
df		Grid	16.00	17.00	14.00	13.00	13.00	10.00	
	Cube	17.00	17.00	13.00	13.00	13.00	11.00	
Mean difference		Grid	7.29	7.44	7.53	7.57	6.71	6.27	
	Cube	7.61	7.72	7.57	8.00	7.21	6.58	
95% confidence interval of the difference	Lower	Grid	6.30	6.49	6.53	6.42	5.40	4.52	
Cube	6.72	7.00	6.49	6.94	6.17	4.80	
Upper	Grid	8.29	8.40	8.53	8.72	8.02	8.03	
Cube	8.50	8.44	8.65	9.06	8.26	8.37	
Mean		Grid	7.29	7.44	7.53	7.57	6.71	6.27	
	Cube	7.61	7.72	7.57	8.00	7.21	6.58	
Std. deviation		Grid	1.93	1.92	1.81	1.99	2.27	2.61	
	Cube	1.79	1.45	1.87	1.84	1.81	2.81	
Std. error mean		Grid	0.47	0.45	0.47	0.53	0.61	0.79	
	Cube	0.42	0.34	0.50	0.49	0.48	0.81	
Significance/P-Value (α = 0.05)	Grid/Cube	0.00	

Learnability.

The users reported the grid-based system in terms of learnability, an averaged score of 86.11%, whereas the same was reported at 82.56% in the case of the cube-based approach (Table 4). Therefore, the grid-based approach was marginally better than the cube. This could be due to the users’ prior comfort and extensive experience in grid-based systems. The results were further statistically analyzed to find the significance of the results using a t-test. According to the results provided in Table 4, no statistical discrepancies were found, and therefore, the results achieved statistical significance when tested on the confidence interval of 95%.

Table 4 Grid-based and cube-based statistical analysis for the QUIS (Learn.) questionnaire.

Statistical test for learn.	Q/System	Q1	Q2	Q3	Q4	Q5	Q6	
t		Grid	18.26	15.22	13.98	17.18	11.46	13.45	
	Cube	17.48	12.87	13.49	17.81	8.02	10.82	
df		Grid	16.00	12.00	16.00	16.00	11.00	10.00	
	Cube	16.00	14.00	16.00	16.00	13.00	10.00	
Mean difference		Grid	8.12	7.62	7.59	7.65	7.17	7.27	
	Cube	7.82	7.27	7.29	7.82	6.50	7.00	
95% confidence interval of the difference	Lower	Grid	7.18	6.52	6.44	6.70	5.79	6.07	
Cube	6.87	6.06	6.15	6.89	4.75	5.56	
Upper	Grid	9.06	8.71	8.74	8.59	8.54	8.48	
Cube	8.77	8.48	8.44	8.75	8.25	8.44	
Mean		Grid	8.12	7.62	7.59	7.65	7.17	7.27	
	Cube	7.82	7.27	7.29	7.82	6.50	7.00	
Std. deviation		Grid	1.83	1.80	2.24	1.84	2.17	1.79	
	Cube	1.85	2.19	2.23	1.81	3.03	2.14	
Std. error mean		Grid	0.44	0.50	0.54	0.45	0.63	0.54	
	Cube	0.45	0.56	0.54	0.44	0.81	0.65	
Significance/P-Value (α = 0.05)	Grid/Cube	0.00	

System capabilities.

The system capability measured the users’ response regarding system accuracy, reliability, flexibility, and adaptability. In this regard, the users reported the grid-based system an average score of 80.78%, whereas the same was reported at 82.78% in the cube-based approach. Therefore, the cube-based approach was marginally better than the grid-based image browsing. The results were further statistically analyzed to find the significance of the results using a t-test. According to the results provided in Table 5, no statistical discrepancies were found, and therefore, the results achieved statistical significance when tested on the confidence interval of 95%.

Table 5 Grid-based and cube-based statistical analysis for the QUIS (Sys.) questionnaire.

Statistical test for sys.	Q/System	Q1	Q2	Q3	Q4	Q5	
t		Grid	11.75	11.94	11.46	11.64	13.42	
	Cube	17.18	25.66	20.54	14.79	23.37	
df		Grid	16.00	14.00	15.00	14.00	16.00	
	Cube	16.00	14.00	15.00	15.00	16.00	
Mean difference		Grid	6.94	7.20	7.19	7.33	7.24	
	Cube	7.53	7.87	7.50	7.63	7.82	
95% confidence interval of the difference	Lower	Grid	5.69	5.91	5.85	5.98	6.09	
Cube	6.60	7.21	6.72	6.53	7.11	
Upper	Grid	8.19	8.49	8.52	8.68	8.38	
Cube	8.46	8.52	8.28	8.72	8.53	
Mean		Grid	6.94	7.20	7.19	7.33	7.24	
	Cube	7.53	7.87	7.50	7.63	7.82	
Std. deviation		Grid	2.44	2.34	2.51	2.44	2.22	
	Cube	1.81	1.19	1.46	2.06	1.38	
Std. error mean		Grid	0.59	0.60	0.63	0.63	0.54	
	Cube	0.44	0.31	0.37	0.52	0.33	
Significance/P-Value (α = 0.05)	Grid/Cube	0.00	

Overall reaction.

The overall reaction category of the QUIS measured the use of first impressions when using the tool, including perceived interest, difficulty, and satisfaction. The users reported the grid-based system an averaged impression score of 84.11%, whereas the same was reported at 81.89% in the case of the cube-based approach (Table 6). Therefore, the users of the grid-based system were found to be more appealing to the grid-based approach than the cube-based image browsing. The results were further statistically analyzed to find the significance of the results using a t-test. According to the results provided in Table 6, no statistical discrepancies were found; therefore, the results achieved statistical significance when tested on the confidence interval of 95%.

Table 6 Grid-based and cube-based statistical analysis for the QUIS (Overall) questionnaire.

Statistical test for overall	Q/System	Q1	Q2	Q3	Q4	Q5	Q6	
t		Grid	14.89	25.67	16.13	14.29	15.37	10.61	
	Cube	14.89	25.67	16.13	14.29	15.37	10.61	
df		Grid	17	15	16	14	16	16	
	Cube	17	15	16	14	16	16	
Mean difference		Grid	7.44	8.44	7.76	7.33	7.47	7.18	
	Cube	7.44	8.44	7.76	7.33	7.47	7.18	
95% confidence interval of the difference	Lower	Grid	6.39	7.74	6.74	6.23	6.44	5.74	
Cube	6.39	7.74	6.74	6.23	6.44	5.74	
Upper	Grid	8.5	9.14	8.79	8.43	8.5	8.61	
Cube	8.5	9.14	8.79	8.43	8.5	8.61	
Mean		Grid	7.44	8.44	7.76	7.33	7.47	7.18	
	Cube	7.44	8.44	7.76	7.33	7.47	7.18	
Std. deviation		Grid	2.12	1.31	1.99	1.99	2	2.79	
	Cube	2.12	1.31	1.99	1.99	2	2.79	
Std. error mean		Grid	0.5	0.33	0.48	0.51	0.49	0.68	
	Cube	0.5	0.33	0.48	0.51	0.49	0.68	
Significance/P-Value (α = 0.05)	Grid/Cube	0.00	

Discussion & Comparison

The mass growth in multimedia data has created challenges in visualizing large datasets over the past few years (Miranda et al., 2017). Most techniques usually rely on some traditional methods of data reduction such as image clustering (Mousavirad, Ebrahimpour-Komleh & Schaefer, 2019), hierarchical browsing (Kovalčík et al., 2020), graph browsing (Barthel, Hezel & Jung, 2017), spherical browsing (Schaefer, Budnik & Krawczyk, 2017), and grid-based (Langenkämper et al., 2017) organization of images. However, a research gap existed in investigating cube-based 3D browsing of images. The web image search results on the SERPs are usually organized via primitive 2-dimensional grid layouts that hinder the in-depth browsing of image results, navigation within the search results, and reachability.

To answer our (RQ1), we represented the web image search results in cubic graphs and enabled their exploration via the instantiated SUI design. Furthermore, there existed a research gap (RQ2) that facilitated the reachability and navigation of the results at lower-ranking positions. To address this issue, the proposed approach further explored a cubic graph space of web image search results via various components of a comprehensive SUI design.

To answer our (RQ3), we evaluated the cubic approach against a traditional grid-based image search engine using QUIS and SUS questionnaires. Mainly, six major parameters were studied: system usability, ease of use, learnability, system capabilities, and overall reaction to the tool. The achieved results showed improvements in screen design, information & terminology, and system capability in the cubic image results exploration over the grid-based image systems.

On the contrary, users rated the overall reaction and system usability of the traditional grid-based image exploration (81.9 & 80.8, respectively) slightly higher than the cubic exploration (84.1 & 78.2, respectively). Similarly, the learnability of the tool was also reported higher on the grid-based system (82.5) over the cubic exploration (86.1). This could be due to users’ need for previous experience in three-dimensional image exploration tools. However, despite a lack of user experience in cubic exploration, the proposed tool is comparable with a marginal difference. This shows the potential of a multidimensional image exploration tool to support in-depth browsing and navigation. These results are further visualized in Fig. 7.

In the I-Cubid approach, we allowed the users to express their visual information needs by introducing simple textual query terms (usually bi-grams and tri-grams) to interact with image search results via multiple cube-based in-depth integrated layouts. The objective was to enable the nonlinear exploration of image results in a usable way. The image search results-based 3D cubic representation comprised multiple planes and the browsing pattern to facilitate the image’s traversing and plane’s switching.

To confirm the efficacy of the I-Cubid approach, we performed a usability analysis, including system usability and user interface satisfaction. Table A4 and Table A5 shows the detailed scores obtained from the users for the traditional grid-based and I-Cubid approach. Similarly, the user interface satisfaction detailed scores report is provided in Table A6 and Table A7, respectively.

Figure 7 Overall average scores for the QUIS and SUS, where the higher score demonstrates better performance.

Mainly, the users reported the system usability mildly higher in the grid-based approach (80.8) than in the cube-based approach (78.2). The lower score for the cube representation can be due to users’ unfamiliarity with the cube-based approach. Almost similar results were reported from the user interface satisfaction usability (82.39%) for the cube-based approach and (82.60%) for the grid-based approach. To answer our (RQ4), according to the obtained results, the margin was minimal despite the said challenges, therefore, showing potential for future improvements.

Conclusion & Future Work

In this research, we started discussing the proliferation utilization of images on the web. We mainly formulated RQ1 that raised the issue of linear web image search results representation over SERPs, limiting the in-depth browsing. To overcome these limitations, we proposed the I-Cubid novel cubic presentation approach that retrieved, visualized, and in-depth browsed the retrieved web image search results from the online web source. Furthermore, there existed a research gap (RQ2) that facilitated the reachability and navigation of the results at lower-ranking positions. To address this issue, the proposed approach further explored a cubic graph space of web image search results via various components of a comprehensive SUI design. To confirm the efficacy of the proposed approach, we formulated RQ3.

To determine whether the in-depth browsing provided via the I-Cubid approach is usable and satisfies the user’s information needs (RQ3), we evaluated the cubic approach against a traditional grid-based image search engine using QUIS and SUS questionnaires. Mainly, six major parameters were studied: system usability, ease of use, learnability, system capabilities, and overall reaction to the tool. The achieved results showed improvements in screen design, information & terminology, and system capability in the cubic image results exploration over the grid-based image systems.

Finally, to unveil whether users prefer the interaction as visualization and in-depth browsing via cubes in the I-Cubid, we performed a usability analysis, including system usability and user interface satisfaction. Mainly, the users reported the system usability mildly higher in the grid-based approach (80.8) than in the cube-based approach (78.2). The lower score for the cube representation can be due to users’ unfamiliarity with the cube-based approach. Almost similar results were reported from the user interface satisfaction usability (82.39%) for the cube-based approach and (82.60%) for the grid-based approach. To answer our (RQ4), according to the obtained results, the margin was minimal despite the said challenges, therefore, showing potential for future improvements.

The usability analysis administered on users with diverse backgrounds yielded cubic improvements in screen design, information & technology, and system capabilities. In contrast, the tool’s system usability, learnability, and reaction were marginally better in the grid-based browsing of web image search results. The obtained results were statistically evaluated and contained statistical significance.

Despite the users’ need for familiarity with the proposed approach, the results show a promising baseline to enhance future web image search results exploration. Therefore, in the future, we are interested in further investigating deep learning approaches to augment recommendation and personalization parameters to aid the cubic browsing of the web image search results. We are further interested in extending the cubic image exploration approach to hyper-cubic video exploration to ease the nonlinear navigation of video content.

Supplemental Information

Supplemental Information 1 User consent form, demographic details, evaluation tasks detail, raw SUS and QUIS responses

Click here for additional data file.

Supplemental Information 2 Raw usability scores, questionnaire scales, and demonstration

The raw SUS and QUIS usability scores for the Cube and Grid interfaces. This also includes the detailed demonstration view, search tasks and queries used for experimental purposes.

Click here for additional data file.

The authors would like to acknowledge the Flickr API for providing images for testing and evaluation purposes from the Flickr search engine.

Additional Information and Declarations

Competing Interests

Author Contributions

Ethics

Data Deposition

The authors declare that there are no competing interests.

Umer Rashid conceived and designed the experiments, performed the experiments, authored or reviewed drafts of the article, and approved the final draft.

Maha Saddal analyzed the data, prepared figures and/or tables, authored or reviewed drafts of the article, and approved the final draft.

Abdur Rehman Khan analyzed the data, prepared figures and/or tables, authored or reviewed drafts of the article, and approved the final draft.

Sadia Manzoor conceived and designed the experiments, performed the experiments, performed the computation work, prepared figures and/or tables, and approved the final draft.

Naveed Ahmad analyzed the data, authored or reviewed drafts of the article, and approved the final draft.

The following information was supplied relating to ethical approvals (i.e., approving body and any reference numbers):

Quaid-i-Azam University Islamabad granted Ethical approval to conduct the study within its facilities vide (Ethical Application Ref: UR \BSCS \Projects \0039).

The following information was supplied regarding data availability:

The raw measurements, evaluation questionnaires, search tasks, and queries, are available in the Supplemental File.

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
