# Peer review of "I-Cubid: a nonlinear cubic graph-based approach to visualize and in-depth browse Flickr image results"

_PeerJ Computer Science, doi:10.7717/peerj-cs.1476_

## Round 0.1 · original submission · Minor Revisions

Please address all noted concerns of the peer reviewers where applicable. If revising and resubmitting, expect further review prior to any forthcoming decision.

Reviewer 1 ·

Basic reporting

I. Overall, this is a well-written article with significant contributions. However, the presentation can be further improved by presenting the text in a logically connected manner. A careful proofreading will help identify and remove the grammatical mistakes and spelling slip-ups.

II. The manuscript structure and its elements such as tables, and figures etc., are professionally structured. The results are well argued, discussed and formulated.

III. The research gaps can be explained better in the light of latest review articles. Some notable examples include:

a. Praveen K. Kopalle, Manish Gangwar, Andreas Kaplan, Divya Ramachandran, Werner Reinartz, Aric Rindfleisch, Examining artificial intelligence (AI) technologies in marketing via a global lens: Current trends and future research opportunities, International Journal of Research in Marketing, Volume 39, Issue 2, 2022, Pages 522-540, https://doi.org/10.1016/j.ijresmar.2021.11.002.

b. Al-Jubouri, H. A. (2019). Content-based image retrieval: survey. J. Eng. Sustainable Dev., 23, 42-63.

c. Alzu’bi, Ahmad, Abbes Amira, and Naeem Ramzan. "Semantic content-based image retrieval: A comprehensive study." Journal of Visual Communication and Image Representation 32 (2015): 20-54.

d. Bouchakwa, Mariam, Yassine Ayadi, and Ikram Amous. "A review on visual content-based and users’ tags-based image annotation: methods and techniques." Multimedia Tools and Applications 79 (2020): 21679-21741.

e. Tekli, Joe. "An overview of cluster-based image search result organization: background, techniques, and ongoing challenges." Knowledge and Information Systems 64.3 (2022): 589-642.

Experimental design

I. The manuscript presents original primary research that fits the aims and scope of the journal. It addresses the issues in traditional image search systems that present a bi-direction search grid on their SERPs, which is extended to a three-dimensional web image search space via the non-linearity in searching, browsing, and navigating images. This research work also tests the usability of its proposed solution of cubic representation over grid layouts.

II. The research questions are clear, relevant, and meaningful and answer the identified knowledge gap.

III. The methods, experimental settings, and results are well-presented and explained.

Validity of the findings

I. The findings are well-presented and valid regarding impact, novelty, and about the gaps in the literature. The underlying data has been provided in a supplementary attachment, which is clear and statistically sound.

II. The conclusion section needs additional work. First, it should summarize the main findings and research implications. Next, it should be linked with the research questions raised in the Introduction section and the gaps in the literature. The limitations should be elaborated further, and future work on addressing them should be provided.

Reviewer 2 ·

Basic reporting

The text is clear and easy to follow. Here are some minor errors spotted.
L45 - researchers
L136 - employ
L202 - nodes
L387 - within-study -> within-subjects
L440-L441, L448-L449 - "as per user's response" - not required.
Many places "users'" is used like L31, L376, L385, L400, L484 whereas many places "user's" is
used like L427, L435, L441, L449, L469. This needs to checked and consistently written.

Good Literature review and proper context is provided.

Figures:
Fig 5,7 - Brief description about the figure should be provided in figure caption
Fig 6 - A rudimentary use case with cats is demonstrated. The authors should demonstrate a
proper uses case that necessitates the need for cubic representation
- The flow of interaction should also be demonstrated mapping it to previous figures
- Current figure has some wasted space which can be utilized for above.
Fig 8 - For different metric if higher or lower is better should be specified in the figure.

Experimental design

Research question are well-defined.

Here are some of the problems or missing details:
- L234-L235 describes the data is queried from Flicker. But not much information is provided if the images are download from Flicker online as the user queries or they are already pre-downloaded. Also, it seems the whole system is very dependent on structured data provided by Flicker. It will better also if the authors show similar search capabilities on unstructured datasets.
- There is no complexity analysis provided.
- The significance of different 'planes' are not clear. As shown in Fig 1, products like Flicker, Google at least provides semantically meaningful 'clusters' or 'facets'. It's not clear what the different 'planes' means even from the demo provided.
- bi-gram/ tri-gram language model is used and Jaccard Similarity is used for matching. While it good as an experimental setting there are much better models available, like glove[1], word2vec[2] and even multimodal embedding space like CLIP[3]

[1] - Pennington et al. "Glove: Global vectors for word representation." Proceedings of the 2014 conference on empirical methods in natural language processing (EMNLP). 2014.
[2] - Mikolov, Tomas, et al. "Efficient estimation of word representations in vector space." arXiv preprint arXiv:1301.3781 (2013).
[3] - https://openai.com/research/clip

Validity of the findings

In the tables what is the significance, is it p-value. Also, what is the set alpha?

Additional comments

Very little system details are provided. L374 goes into a bit, but more relevant details are missing. Like whether the SUI is developed as a webservice or it's a standalone application. From the demo its clear its a webservice but in the text it's not mentioned. Not much information is provided how the data is handled and hosted.

---

## Round 0.2 · accepted · Accept

Authors have made appropriate changes as per reviewers. Thank you.

Reviewer 1 ·

Basic reporting

This a well-written and well-presented research contribution to the body of knowledge on image search. The language is clear, and easy to comprehend, with sufficient and up-to-date references to the recent relevant literature. The results are clear and support the research questions/hypotheses raised in this manuscript.

Experimental design

The experimental work is original with sufficient details and formalism. All the relevant results are present.

Validity of the findings

The manuscript presents original primary research that fits the aims and scope of the journal. It addresses the issues in traditional image search systems that present a bi-direction search grid on their SERPs, which is extended to a three-dimensional web image search space via the non-linearity in searching, browsing, and navigating images. This research work also tests the usability of its proposed solution of cubic representation over grid layouts.

Additional comments

The authors have well-revised the manuscript in the light of my suggested revisions.

Reviewer 2 ·

Basic reporting

The authors have addressed the concerns and have updated the text accordingly.

Experimental design

The authors have addressed the concerns and have updated the text accordingly.

Validity of the findings

The authors have addressed the concerns and have updated the text accordingly.